# Information-theoretic analysis of generalization capability of learning algorithms

**Aolin Xu**     **Maxim Raginsky**
{aolinxu2,maxim}@illinois.edu *

## Abstract

We derive upper bounds on the generalization error of a learning algorithm in terms of the mutual information between its input and output. The bounds provide an information-theoretic understanding of generalization in learning problems, and give theoretical guidelines for striking the right balance between data fit and generalization by controlling the input-output mutual information. We propose a number of methods for this purpose, among which are algorithms that regularize the ERM algorithm with relative entropy or with random noise. Our work extends and leads to nontrivial improvements on the recent results of Russo and Zou.

## 1   Introduction

A learning algorithm can be viewed as a randomized mapping, or a channel in the information-theoretic language, which takes a training dataset as input and generates a hypothesis as output. The generalization error is the difference between the population risk of the output hypothesis and its empirical risk on the training data. It measures how much the learned hypothesis suffers from overfitting. The traditional way of analyzing the generalization error relies either on certain complexity measures of the hypothesis space, e.g. the VC dimension and the Rademacher complexity [1], or on certain properties of the learning algorithm, e.g., uniform stability [2]. Recently, motivated by improving the accuracy of adaptive data analysis, Russo and Zou [3] showed that the mutual information between the collection of empirical risks of the available hypotheses and the final output of the algorithm can be used effectively to analyze and control the bias in data analysis, which is equivalent to the generalization error in learning problems. Compared to the methods of analysis based on differential privacy, e.g., by Dwork et al. [4,5] and Bassily et al. [6], the method proposed in [3] is simpler and can handle unbounded loss functions; moreover, it provides elegant information-theoretic insights into improving the generalization capability of learning algorithms. In a similar information-theoretic spirit, Alabdulmohsin [7, 8] proposed to bound the generalization error in learning problems using the total-variation information between a random instance in the dataset and the output hypothesis, but the analysis apply only to bounded loss functions.

In this paper, we follow the information-theoretic framework proposed by Russo and Zou [3] to derive upper bounds on the generalization error of learning algorithms. We extend the results in [3] to the situation where the hypothesis space is uncountably infinite, and provide improved upper bounds on the expected absolute generalization error. We also obtain concentration inequalities for the generalization error, which were not given in [3]. While the main quantity examined in [3] is the mutual information between the collection of empirical risks of the hypotheses and the output of the algorithm, we mainly focus on relating the generalization error to the mutual information between the input dataset and the output of the algorithm, which formalizes the intuition that the less information

a learning algorithm can extract from the input dataset, the less it will overfit. This viewpoint provides theoretical guidelines for striking the right balance between data fit and generalization by controlling the algorithm's input-output mutual information. For example, we show that regularizing the empirical risk minimization (ERM) algorithm with the input-output mutual information leads to the well-known Gibbs algorithm. As another example, regularizing the ERM algorithm with random noise can also control the input-output mutual information. For both the Gibbs algorithm and the noisy ERM algorithm, we also discuss how to calibrate the regularization in order to incorporate any prior knowledge of the population risks of the hypotheses into algorithm design. Additionally, we discuss adaptive composition of learning algorithms, and show that the generalization capability of the overall algorithm can be analyzed by examining the input-output mutual information of the constituent algorithms.

Another advantage of relating the generalization error to the input-output mutual information is that the latter quantity depends on all ingredients of the learning problem, including the distribution of the dataset, the hypothesis space, the learning algorithm itself, and potentially the loss function, in contrast to the VC dimension or the uniform stability, which only depend on the hypothesis space or on the learning algorithm. As the generalization error can strongly depend on the input dataset [9], the input-output mutual information can be more tightly coupled to the generalization error than the traditional generalization-guaranteeing quantities of interest. We hope that our work can provide some information-theoretic understanding of generalization in modern learning problems, which may not be sufficiently addressed by the traditional analysis tools [9].

For the rest of this section, we define the quantities that will be used in the paper. In the standard framework of statistical learning theory [10], there is an instance space $\mathsf{Z}$, a hypothesis space $\mathsf{W}$, and a nonnegative loss function $\ell : \mathsf{W} \times \mathsf{Z} \to \mathbb{R}^+$. A learning algorithm characterized by a Markov kernel $P_{W|S}$ takes as input a dataset of size $n$, i.e., an $n$-tuple

$$S = (Z_1, \ldots, Z_n) \tag{1}$$

of i.i.d. random elements of $\mathsf{Z}$ with some unknown distribution $\mu$, and picks a random element $W$ of $\mathsf{W}$ as the output hypothesis according to $P_{W|S}$. The population risk of a hypothesis $w \in \mathsf{W}$ on $\mu$ is

$$L_\mu(w) \triangleq \mathbb{E}[\ell(w, Z)] = \int_{\mathsf{Z}} \ell(w, z)\mu(\mathrm{d}z). \tag{2}$$

The goal of learning is to ensure that the population risk of the output hypothesis $W$ is small, either in expectation or with high probability, under any data generating distribution $\mu$. The excess risk of $W$ is the difference $L_\mu(W) - \inf_{w \in \mathsf{W}} L_\mu(w)$, and its expected value is denoted as $R_{\mathrm{excess}}(\mu, P_{W|S})$. Since $\mu$ is unknown, the learning algorithm cannot directly compute $L_\mu(w)$ for any $w \in \mathsf{W}$, but can instead compute the empirical risk of $w$ on the dataset $S$ as a proxy, defined as

$$L_S(w) \triangleq \frac{1}{n} \sum_{i=1}^n \ell(w, Z_i). \tag{3}$$

For a learning algorithm characterized by $P_{W|S}$, the generalization error on $\mu$ is the difference $L_\mu(W) - L_S(W)$, and its expected value is denoted as

$$\mathrm{gen}(\mu, P_{W|S}) \triangleq \mathbb{E}[L_\mu(W) - L_S(W)], \tag{4}$$

where the expectation is taken with respect to the joint distribution $P_{S,W} = \mu^{\otimes n} \otimes P_{W|S}$. The expected population risk can then be decomposed as

$$\mathbb{E}[L_\mu(W)] = \mathbb{E}[L_S(W)] + \mathrm{gen}(\mu, P_{W|S}), \tag{5}$$

where the first term reflects how well the output hypothesis fits the dataset, while the second term reflects how well the output hypothesis generalizes. To minimize $\mathbb{E}[L_\mu(W)]$ we need both terms in (5) to be small. However, it is generally impossible to minimize the two terms simultaneously, and any learning algorithm faces a trade-off between the empirical risk and the generalization error. In what follows, we will show how the generalization error can be related to the mutual information between the input and output of the learning algorithm, and how we can use these relationships to guide the algorithm design to reduce the population risk by balancing fitting and generalization.

## 2 Algorithmic stability in input-output mutual information

As discussed above, having a small generalization error is crucial for a learning algorithm to produce an output hypothesis with a small population risk. It turns out that the generalization error of a learning algorithm can be determined by its stability properties. Traditionally, a learning algorithm is said to be stable if a small change of the input to the algorithm does not change the output of the algorithm much. Examples include uniform stability defined by Bousquet and Elisseeff [2] and on-average stability defined by Shalev-Shwartz et al. [11]. In recent years, information-theoretic stability notions, such as those measured by differential privacy [5], KL divergence [6,12], total-variation information [7], and erasure mutual information [13], have been proposed. All existing notions of stability show that the generalization capability of a learning algorithm hinges on how sensitive the output of the algorithm is to local modifications of the input dataset. It implies that the less dependent the output hypothesis $W$ is on the input dataset $S$, the better the learning algorithm generalizes. From an information-theoretic point of view, the dependence between $S$ and $W$ can be naturally measured by the mutual information between them, which prompts the following information-theoretic definition of stability. We say that a learning algorithm is $(\varepsilon, \mu)$-stable in input-output mutual information if, under the data-generating distribution $\mu$,

$$I(S; W) \leq \varepsilon. \tag{6}$$

Further, we say that a learning algorithm is $\varepsilon$-stable in input-output mutual information if

$$\sup_{\mu} I(S; W) \leq \varepsilon. \tag{7}$$

According to the definitions in (6) and (7), the less information the output of a learning algorithm can provide about its input dataset, the more stable it is. Interestingly, if we view the learning algorithm $P_{W|S}$ as a channel from $\mathsf{Z}^n$ to $\mathsf{W}$, the quantity $\sup_{\mu} I(S; W)$ can be viewed as the information capacity of the channel, under the constraint that the input distribution is of a product form. The definition in (7) means that a learning algorithm is more stable if its information capacity is smaller. The advantage of the weaker definition in (6) is that $I(S; W)$ depends on both the algorithm and the distribution of the dataset. Therefore, it can be more tightly coupled with the generalization error, which itself depends on the dataset. We mainly focus on studying the consequence of this notion of $(\varepsilon, \mu)$-stability in input-output mutual information for the rest of this paper.

## 3 Upper-bounding generalization error via $I(S; W)$

In this section, we derive various generalization guarantees for learning algorithms that are stable in input-output mutual information.

### 3.1 A decoupling estimate

We start with a digression from the statistical learning problem to a more general problem, which may be of independent interest. Consider a pair of random variables $X$ and $Y$ with joint distribution $P_{X,Y}$. Let $\bar{X}$ be an independent copy of $X$, and $\bar{Y}$ an independent copy of $Y$, such that $P_{\bar{X},\bar{Y}} = P_X \otimes P_Y$. For an arbitrary real-valued function $f : \mathsf{X} \times \mathsf{Y} \to \mathbb{R}$, we have the following upper bound on the absolute difference between $\mathbb{E}[f(X, Y)]$ and $\mathbb{E}[f(\bar{X}, \bar{Y})]$.

**Lemma 1** (proved in Appendix A). *If $f(\bar{X}, \bar{Y})$ is $\sigma$-subgaussian under $P_{\bar{X},\bar{Y}} = P_X \otimes P_Y$ [2] , then*

$$\left| \mathbb{E}[f(X, Y)] - \mathbb{E}[f(\bar{X}, \bar{Y})] \right| \leq \sqrt{2\sigma^2 I(X; Y)}. \tag{8}$$

### 3.2 Upper bound on expected generalization error

Upper-bounding the generalization error of a learning algorithm $P_{W|S}$ can be cast as a special case of the preceding problem, by setting $X = S, Y = W$, and $f(s, w) = \frac{1}{n} \sum_{i=1}^{n} \ell(w, z_i)$. For an arbitrary $w \in \mathsf{W}$, the empirical risk can be expressed as $L_S(w) = f(S, w)$ and the population risk can be expressed as $L_\mu(w) = \mathbb{E}[f(S, w)]$. Moreover, the expected generalization error can be written as

$$\mathrm{gen}(\mu, P_{W|S}) = \mathbb{E}[f(\bar{S}, \bar{W})] - \mathbb{E}[f(S, W)], \tag{9}$$

where the joint distribution of $S$ and $W$ is $P_{S,W} = \mu^{\otimes n} \otimes P_{W|S}$. If $\ell(w, Z)$ is $\sigma$-subgaussian for all $w \in \mathsf{W}$, then $f(S, w)$ is $\sigma/\sqrt{n}$-subgaussian due to the i.i.d. assumption on $Z_i$'s, hence $f(\bar{S}, \bar{W})$ is $\sigma/\sqrt{n}$-subgaussian. This, together with Lemma 1, leads to the following theorem.

**Theorem 1.** *Suppose $\ell(w, Z)$ is $\sigma$-subgaussian under $\mu$ for all $w \in \mathsf{W}$, then*

$$\left| \text{gen}(\mu, P_{W|S}) \right| \leq \sqrt{\frac{2\sigma^2}{n} I(S; W)}. \tag{10}$$

Theorem 1 suggests that, by controlling the mutual information between the input and the output of a learning algorithm, we can control its generalization error. The theorem allows us to consider unbounded loss functions as long as the subgaussian condition is satisfied. For a bounded loss function $\ell(\cdot, \cdot) \in [a, b]$, $\ell(w, Z)$ is guaranteed to be $(b - a)/2$-subgaussian for all $\mu$ and all $w \in \mathsf{W}$.

Russo and Zou [3] considered the same problem setup with the restriction that the hypothesis space $\mathsf{W}$ is finite, and showed that $|\text{gen}(\mu, P_{W|S})|$ can be upper-bounded in terms of $I(\Lambda_\mathsf{W}(S); W)$, where

$$\Lambda_\mathsf{W}(S) \triangleq \left( L_S(w) \right)_{w \in \mathsf{W}} \tag{11}$$

is the collection of empirical risks of the hypotheses in $\mathsf{W}$. Using Lemma 1 by setting $X = \Lambda_\mathsf{W}(S)$, $Y = W$, and $f(\Lambda_\mathsf{W}(s), w) = L_s(w)$, we immediately recover the result by Russo and Zou even when $\mathsf{W}$ is uncountably infinite:

**Theorem 2** (Russo and Zou [3]). *Suppose $\ell(w, Z)$ is $\sigma$-subgaussian under $\mu$ for all $w \in \mathsf{W}$, then*

$$\left| \text{gen}(\mu, P_{W|S}) \right| \leq \sqrt{\frac{2\sigma^2}{n} I(\Lambda_\mathsf{W}(S); W)}. \tag{12}$$

It should be noted that Theorem 1 can be obtained as a consequence of Theorem 2 because

$$I(\Lambda_\mathsf{W}(S); W) \leq I(S; W), \tag{13}$$

which is due to the Markov chain $\Lambda_\mathsf{W}(S) - S - W$, as for each $w \in \mathsf{W}$, $L_S(w)$ is a function of $S$. However, if the output $W$ depends on $S$ only through the empirical risks $\Lambda_\mathsf{W}(S)$, in other words, when the Markov chain $S - \Lambda_\mathsf{W}(S) - W$ holds, then Theorem 1 and Theorem 2 are equivalent. The advantage of Theorem 1 is that $I(S; W)$ can be much easier to evaluate than $I(\Lambda_\mathsf{W}(S); W)$, and can provide better insights to guide the algorithm design. We will elaborate on this when we discuss the Gibbs algorithm and the adaptive composition of learning algorithms.

Theorem 1 and Theorem 2 only provide upper bounds on the expected generalization error. We are often interested in analyzing the absolute generalization error $|L_\mu(W) - L_S(W)|$, e.g., its expected value or the probability for it to be small. We need to develop stronger tools to tackle these problems, which is the subject of the next two subsections.

### 3.3 A concentration inequality for $|L_\mu(W) - L_S(W)|$

For any fixed $w \in \mathsf{W}$, if $\ell(w, Z)$ is $\sigma$-subgaussian, the Chernoff-Hoeffding bound gives $\mathbb{P}[|L_\mu(w) - L_S(w)| > \alpha] \leq 2e^{-\alpha^2 n/2\sigma^2}$. It implies that, if $S$ and $W$ are independent, then a sample size of

$$n = \frac{2\sigma^2}{\alpha^2} \log \frac{2}{\beta} \tag{14}$$

suffices to guarantee

$$\mathbb{P}[|L_\mu(W) - L_S(W)| > \alpha] \leq \beta. \tag{15}$$

The following results show that, when $W$ is dependent on $S$, as long as $I(S; W)$ is sufficiently small, a sample complexity polynomial in $1/\alpha$ and logarithmic in $1/\beta$ still suffices to guarantee (15), where the probability now is taken with respect to the joint distribution $P_{S,W} = \mu^{\otimes n} \otimes P_{W|S}$.

**Theorem 3** (proved in Appendix B). *Suppose $\ell(w, Z)$ is $\sigma$-subgaussian under $\mu$ for all $w \in \mathsf{W}$. If a learning algorithm satisfies $I(\Lambda_\mathsf{W}(S); W) \leq \varepsilon$, then for any $\alpha > 0$ and $0 < \beta \leq 1$, (15) can be guaranteed by a sample complexity of*

$$n = \frac{8\sigma^2}{\alpha^2} \left( \frac{\varepsilon}{\beta} + \log \frac{2}{\beta} \right). \tag{16}$$

In view of (13), any learning algorithm that is $(\varepsilon, \mu)$-stable in input-output mutual information satisfies the condition $I(\Lambda_{\mathsf{W}}(S); W) \leq \varepsilon$. The proof of Theorem 3 is based on Lemma 1 and an adaptation of the "monitor technique" proposed by Bassily et al. [6]. While the high-probability bounds of [4–6] based on differential privacy are for bounded loss functions and for functions with bounded differences, the result in Theorem 3 only requires $\ell(w, Z)$ to be subgaussian. We have the following corollary of Theorem 3.

**Corollary 1.** *Under the conditions in Theorem 3, if for some function $g(n) \geq 1$, $\varepsilon \leq (g(n) - 1)\beta \log \frac{2}{\beta}$, then a sample complexity that satisfies $n/g(n) \geq \frac{8\sigma^2}{\alpha^2} \log \frac{2}{\beta}$ guarantees (15).*

For example, taking $g(n) = 2$, Corollary 1 implies that if $\varepsilon \leq \beta \log(2/\beta)$, then (15) can be guaranteed by a sample complexity of $n = (16\sigma^2/\alpha^2) \log(2/\beta)$, which is on the same order of the sample complexity when $S$ and $W$ are independent as in (14). As another example, taking $g(n) = \sqrt{n}$, Corollary 1 implies that if $\varepsilon \leq (\sqrt{n} - 1)\beta \log(2/\beta)$, then a sample complexity of $n = (64\sigma^4/\alpha^4)(\log(2/\beta))^2$ guarantees (15).

### 3.4 Upper bound on $\mathbb{E}|L_\mu(W) - L_S(W)|$

A byproduct of the proof of Theorem 3 (setting $m = 1$ in the proof) is an upper bound on the expected absolute generalization error.

**Theorem 4.** *Suppose $\ell(w, Z)$ is $\sigma$-subgaussian under $\mu$ for all $w \in \mathsf{W}$. If a learning algorithm satisfies that $I(\Lambda_{\mathsf{W}}(S); W) \leq \varepsilon$, then*

$$\mathbb{E}|L_\mu(W) - L_S(W)| \leq \sqrt{\frac{2\sigma^2}{n}(\varepsilon + \log 2)}. \tag{17}$$

This result improves [3, Prop. 3.2], which states that $\mathbb{E}|L_S(W) - L_\mu(W)| \leq \sigma/\sqrt{n} + 36\sqrt{2\sigma^2\varepsilon/n}$. Theorem 4 together with Markov's inequality implies that (15) can be guaranteed by $n = \frac{2\sigma^2}{\alpha^2\beta^2}(\varepsilon + \log 2)$, but it has a worse dependence on $\beta$ as compared to the sample complexity given by Theorem 3.

## 4 Learning algorithms with input-output mutual information stability

In this section, we discuss several learning problems and algorithms from the viewpoint of input-output mutual information stability. We first consider two cases where the input-output mutual information can be upper-bounded via the properties of the hypothesis space. Then we propose two learning algorithms with controlled input-output mutual information by regularizing the ERM algorithm. We also discuss other methods to induce input-output mutual information stability, and the stability of learning algorithms obtained from adaptive composition of constituent algorithms.

### 4.1 Countable hypothesis space

When the hypothesis space is countable, the input-output mutual information can be directly upper-bounded by $H(W)$, the entropy of $W$. If $|\mathsf{W}| = k$, we have $H(W) \leq \log k$. From Theorem 1, if $\ell(w, Z)$ is $\sigma$-subgaussian for all $w \in \mathsf{W}$, then for any learning algorithm $P_{W|S}$ with countable $\mathsf{W}$,

$$|\text{gen}(\mu, P_{W|S})| \leq \sqrt{\frac{2\sigma^2 H(W)}{n}}. \tag{18}$$

For the ERM algorithm, the upper bounds for the expected generalization error also hold for the expected excess risk, since the empirical risk of the ERM algorithm satisfies

$$\mathbb{E}[L_S(W_{\text{ERM}})] = \mathbb{E}\left[\inf_{w \in \mathsf{W}} L_S(w)\right] \leq \inf_{w \in \mathsf{W}} \mathbb{E}[L_S(w)] = \inf_{w \in \mathsf{W}} L_\mu(w). \tag{19}$$

For an uncountable hypothesis space, we can always convert it to a finite one by quantizing the output hypothesis. For example, if $\mathsf{W} \subset \mathbb{R}^m$, we can define the covering number $N(r, \mathsf{W})$ as the cardinality of the smallest set $\mathsf{W}' \subset \mathbb{R}^m$ such that for all $w \in \mathsf{W}$ there is $w' \in \mathsf{W}'$ with $\|w - w'\| \leq r$, and we can use $\mathsf{W}'$ as the codebook for quantization. The final output hypothesis $W'$ will be an element of

W'. If W lies in a $d$-dimensional subspace of $\mathbb{R}^m$ and $\max_{w \in W} \|w\| = B$, then setting $r = 1/\sqrt{n}$, we have $N(r, W) \leq (2B\sqrt{dn})^d$, and under the subgaussian condition of $\ell$,

$$\left|\mathrm{gen}(\mu, P_{W'|S})\right| \leq \sqrt{\frac{2\sigma^2 d}{n} \log\left(2B\sqrt{dn}\right)}. \tag{20}$$

## 4.2 Binary Classification

For the problem of binary classification, $Z = X \times Y$, $Y = \{0, 1\}$, W is a collection of classifiers $w : X \to Y$, which could be uncountably infinite, and $\ell(w, z) = \mathbf{1}\{w(x) \neq y\}$. Using Theorem 1, we can perform a simple analysis of the following two-stage algorithm [14, 15] that can achieve the same performance as ERM. Given the dataset $S$, split it into $S_1$ and $S_2$ with lengths $n_1$ and $n_2$. First, pick a subset of hypotheses $W_1 \subset W$ based on $S_1$ such that $(w(X_1), \ldots, w(X_{n_1}))$ for $w \in W_1$ are all distinct and $\{(w(X_1), \ldots, w(X_{n_1})), w \in W_1\} = \{(w(X_1), \ldots, w(X_{n_1})), w \in W\}$. In other words, $W_1$ forms an empirical cover of W with respect to $S_1$. Then pick a hypothesis from $W_1$ with the minimal empirical risk on $S_2$, i.e.,

$$W = \arg\min_{w \in W_1} L_{S_2}(w). \tag{21}$$

Denoting the $n$th shatter coefficient and the VC dimension of W by $\mathbb{S}_n$ and $V$, we can upper-bound the expected generalization error of $W$ with respect to $S_2$ as

$$\mathbb{E}[L_\mu(W)] - \mathbb{E}[L_{S_2}(W)] = \mathbb{E}\left[\mathbb{E}[L_\mu(W) - L_{S_2}(W)|S_1]\right] \leq \sqrt{\frac{V \log(n_1 + 1)}{2n_2}}, \tag{22}$$

where we have used the fact that $I(S_2; W|S_1 = s_1) \leq H(W|S_1 = s_1) \leq \log \mathbb{S}_{n_1} \leq V \log(n_1 + 1)$, by Sauer's Lemma, and Theorem 1. It can also be shown that [14, 15]

$$\mathbb{E}[L_{S_2}(W)] \leq \mathbb{E}\left[\inf_{w \in W_1} L_\mu(w)\right] \leq \inf_{w \in W} L_\mu(w) + c\sqrt{\frac{V}{n_1}}, \tag{23}$$

where the second expectation is taken with respect to $W_1$ which depends on $S_1$, and $c$ is a constant. Combining (22) and (23) and setting $n_1 = n_2 = n/2$, we have for some constant $c$,

$$\mathbb{E}[L_\mu(W)] \leq \inf_{w \in W} L_\mu(w) + c\sqrt{\frac{V \log n}{n}}. \tag{24}$$

From an information-theoretic point of view, the above two-stage algorithm effectively controls the conditional mutual information $I(S_2; W|S_1)$ by extracting an empirical cover of W using $S_1$, while maintaining a small empirical risk using $S_2$.

## 4.3 Gibbs algorithm

As Theorem 1 shows that the generalization error can be upper-bounded in terms of $I(S; W)$, it is natural to consider an algorithm that minimizes the empirical risk regularized by $I(S; W)$:

$$P_{W|S}^\star = \arg\inf_{P_{W|S}} \left(\mathbb{E}[L_S(W)] + \frac{1}{\beta} I(S; W)\right), \tag{25}$$

where $\beta > 0$ is a parameter that balances fitting and generalization. To deal with the issue that $\mu$ is unknown to the learning algorithm, we can relax the above optimization problem by replacing $I(S; W)$ with an upper bound $D(P_{W|S}\|Q|P_S) = I(S; W) + D(P_W\|Q)$, where $Q$ is an arbitrary distribution on W and $D(P_{W|S}\|Q|P_S) = \int_{Z^n} D(P_{W|S=s}\|Q)\mu^{\otimes n}(\mathrm{d}s)$, so that the solution of the relaxed optimization problem does not depend on $\mu$. It turns out that the well-known Gibbs algorithm solves the relaxed optimization problem.

**Theorem 5** (proved in Appendix C). *The solution to the optimization problem*

$$P_{W|S}^* = \arg\inf_{P_{W|S}} \left(\mathbb{E}[L_S(W)] + \frac{1}{\beta} D(P_{W|S}\|Q|P_S)\right) \tag{26}$$

*is the Gibbs algorithm, which satisfies*

$$P_{W|S=s}^*(\mathrm{d}w) = \frac{e^{-\beta L_s(w)} Q(\mathrm{d}w)}{\mathbb{E}_Q[e^{-\beta L_s(W)}]} \qquad \text{for each } s \in Z^n. \tag{27}$$

We would not have been able to arrive at the Gibbs algorithm had we used $I(\Lambda_W(S); W)$ as the regularization term instead of $I(S; W)$ in (25), even if we upper-bound $I(\Lambda_W(S))$ by $D(P_{W|\Lambda_W(S)}\|Q|P_{\Lambda_W(S)})$. Using the fact that the Gibbs algorithm is $(2\beta/n, 0)$-differentially private when $\ell \in [0, 1]$ [16] and the group property of differential privacy [17], we can upper-bound the input-output mutual information of the Gibbs algorithm as $I(S; W) \leq 2\beta$. Then from Theorem 1, we know that for $\ell \in [0, 1]$, $\left|\text{gen}(\mu, P^*_{W|S})\right| \leq \sqrt{\beta/n}$. Using Hoeffding's lemma, a tighter upper bound on the expected generalization error for the Gibbs algorithm is obtained in [13], which states that if $\ell \in [0, 1]$,

$$\left|\text{gen}(\mu, P^*_{W|S})\right| \leq \frac{\beta}{2n}. \tag{28}$$

With the guarantee on the generalization error, we can analyze the population risk of the Gibbs algorithm. We first present a result for countable hypothesis spaces.

**Corollary 2** (proved in Appendix D). *Suppose* W *is countable. Let* $W$ *denote the output of the Gibbs algorithm applied on dataset S, and let* $w_o$ *denote the hypothesis that achieves the minimum population risk among* W*. For* $\ell \in [0, 1]$*, the population risk of* $W$ *satisfies*

$$\mathbb{E}[L_\mu(W)] \leq \inf_{w\in W} L_\mu(w) + \frac{1}{\beta} \log \frac{1}{Q(w_o)} + \frac{\beta}{2n}. \tag{29}$$

The distribution $Q$ in the Gibbs algorithm can be used to express our preference, or our prior knowledge of the population risks, of the hypotheses in W, in a way that a higher probability under $Q$ is assigned to a hypothesis that we prefer. For example, we can order the hypotheses according to our prior knowledge of their population risks, and set $Q(w_i) = 6/\pi^2 i^2$ for the $i$th hypothesis in the order, then, setting $\beta = \sqrt{n}$, (29) becomes

$$\mathbb{E}[L_\mu(W)] \leq \inf_{w\in W} L_\mu(w) + \frac{2\log i_o + 1}{\sqrt{n}}, \tag{30}$$

where $i_o$ is the index of $w_o$. It means that a better prior knowledge on the population risks leads to a smaller sample complexity to achieve a certain expected excess risk. As another example, if $|W| = k$ and we have no preference on any hypothesis, then taking $Q$ as the uniform distribution on W and setting $\beta = 2\sqrt{n\log k}$, (29) becomes $\mathbb{E}[L_\mu(W)] \leq \inf_{w\in W} L_\mu(w) + \sqrt{(1/n)\log k}$.

For uncountable hypothesis spaces, we can do a similar analysis for the population risk under a Lipschitz assumption on the loss function.

**Corollary 3** (proved in Appendix E). *Suppose* W $= \mathbb{R}^d$*. Let* $w_o$ *be the hypothesis that achieves the minimum population risk among* W*. Suppose* $\ell \in [0, 1]$ *and* $\ell(\cdot, z)$ *is* $\rho$*-Lipschitz for all* $z \in$ Z*. Let* $W$ *denote the output of the Gibbs algorithm applied on dataset S. The population risk of* $W$ *satisfies*

$$\mathbb{E}[L_\mu(W)] \leq \inf_{w\in W} L_\mu(w) + \frac{\beta}{2n} + \inf_{a>0} \left( a\rho\sqrt{d} + \frac{1}{\beta} D(\mathcal{N}(w_o, a^2 \mathbf{I}_d)\|Q) \right). \tag{31}$$

Again, we can use the distribution $Q$ to express our preference of the hypotheses in W. For example, we can choose $Q = \mathcal{N}(w_Q, b^2 \mathbf{I}_d)$ with $b = n^{-1/4}d^{-1/4}\rho^{-1/2}$ and choose $\beta = n^{3/4}d^{1/4}\rho^{1/2}$. Then, setting $a = b$ in (31), we have

$$\mathbb{E}[L_\mu(W)] \leq \inf_{w\in W} L_\mu(w) + \frac{d^{1/4}\rho^{1/2}}{2n^{1/4}} \left( \|w_Q - w_o\|^2 + 3 \right). \tag{32}$$

This result essentially has no restriction on W, which could be unbounded, and only requires the Lipschitz condition on $\ell(\cdot, z)$, which could be non-convex. The sample complexity decreases with a better prior knowledge of the optimal hypothesis.

## 4.4 Noisy empirical risk minimization

Another algorithm with controlled input-output mutual information is the noisy empirical risk minimization algorithm, where independent noise $N_w$, $w \in$ W, is added to the empirical risk of each hypothesis, and the algorithm outputs a hypothesis that minimizes the noisy empirical risks:

$$W = \arg\min_{w\in W} \left( L_S(w) + N_w \right). \tag{33}$$

Similar to the Gibbs algorithm, we can express our preference of the hypotheses by controlling the amount of noise added to each hypothesis, such that our preferred hypotheses will be more likely to be selected when they have similar empirical risks as other hypotheses. The following result formalizes this idea.

**Corollary 4** (proved in Appendix F). *Suppose* $W$ *is countable and is indexed such that a hypothesis with a lower index is preferred over one with a higher index. Also suppose* $\ell \in [0, 1]$. *For the noisy ERM algorithm in* (33), *choosing* $N_i$ *to be an exponential random variable with mean* $b_i$, *we have*

$$\mathbb{E}[L_\mu(W)] \leq \min_i L_\mu(w_i) + b_{i_\circ} + \sqrt{\frac{1}{2n} \sum_{i=1}^{\infty} \frac{L_\mu(w_i)}{b_i} - \left(\sum_{i=1}^{\infty} \frac{1}{b_i}\right)^{-1}}, \tag{34}$$

*where* $i_\circ = \arg\min_i L_\mu(w_i)$. *In particular, choosing* $b_i = i^{1.1}/n^{1/3}$, *we have*

$$\mathbb{E}[L_\mu(W)] \leq \min_i L_\mu(w_i) + \frac{i_\circ^{1.1} + 3}{n^{1/3}}. \tag{35}$$

Without adding noise, the ERM algorithm applied to the above case when $|W| = k$ can achieve $\mathbb{E}[L_\mu(W_{\text{ERM}})] \leq \min_{i \in [k]} L_\mu(w_i) + \sqrt{(1/2n)\log k}$. Compared with (35), we see that performing noisy ERM may be beneficial when we have high-quality prior knowledge of $w_\circ$ and when $k$ is large.

## 4.5   Other methods to induce input-output mutual information stability

In addition to the Gibbs algorithm and the noisy ERM algorithm, many other methods may be used to control the input-output mutual information of the learning algorithm. One method is to preprocess the dataset $S$ to obtain $\tilde{S}$, and then run a learning algorithm on $\tilde{S}$. The preprocessing can be adding noise to the data or erasing some of the instances in the dataset, etc. In any case, we have the Markov chain $S - \tilde{S} - W$, which implies $I(S; W) \leq \min\{I(S; \tilde{S}), I(\tilde{S}; W)\}$. Another method is the postprocessing of the output of a learning algorithm. For example, the weights $\tilde{W}$ generated by a neural network training algorithm can be quantized or perturbed by noise. This gives rise to the Markov chain $S - \tilde{W} - W$, which implies $I(S; W) \leq \min\{I(\tilde{W}; W), I(S; \tilde{W})\}$. Moreover, strong data processing inequalities [18] may be used to sharpen these upper bounds on $I(S; W)$. Preprocessing of the dataset and postprocessing of the output hypothesis are among numerous regularization methods used in the field of deep learning [19, Ch. 7.5]. Other regularization methods may also be interpreted as ways to induce the input-output mutual information stability of a learning algorithm, and this would be an interesting direction of future research.

## 4.6   Adaptive composition of learning algorithms

Beyond analyzing the generalization error of individual learning algorithms, examining the input-output mutual information is also useful for analyzing the generalization capability of complex learning algorithms obtained by adaptively composing simple constituent algorithms. Under a $k$-fold adaptive composition, the dataset $S$ is shared by $k$ learning algorithms that are sequentially executed. For $j = 1, \ldots, k$, the output $W_j$ of the $j$th algorithm may be drawn from a different hypothesis space $W_j$ based on $S$ and the outputs $W^{j-1}$ of the previously executed algorithms, according to $P_{W_j|S,W^{j-1}}$. An example with $k = 2$ is model selection followed by a learning algorithm using the same dataset. Various boosting techniques in machine learning can also be viewed as instances of adaptive composition. From the data processing inequality and the chain rule of mutual information,

$$I(S; W_k) \leq I(S; W^k) = \sum_{j=1}^{k} I(S; W_j | W^{j-1}). \tag{36}$$

If the Markov chain $S - \Lambda_{W_j}(S) - W_j$ holds conditional on $W^{j-1}$ for $j = 1, \ldots, k$, then the upper bound in (36) can be sharpened to $\sum_{j=1}^{k} I(\Lambda_{W_j}(S); W_j | W^{j-1})$. We can thus control the generalization error of the final output by controlling the conditional mutual information at each step of the composition. This also gives us a way to analyze the generalization error of the composed learning algorithm using the knowledge of local generalization guarantees of the constituent algorithms.

## Acknowledgement

We would like to thank Vitaly Feldman and Vivek Bagaria for pointing out errors in the earlier version of this paper. We also would like to thank Peng Guan for helpful discussions.

## Footnotes

*Department of Electrical and Computer Engineering and Coordinated Science Laboratory, University of Illinois, Urbana, IL 61801, USA. This work was supported in part by the NSF CAREER award CCF-1254041 and in part by the Center for Science of Information (CSoI), an NSF Science and Technology Center, under grant agreement CCF-0939370.

[2]Recall that a random variable $U$ is $\sigma$-subgaussian if $\log \mathbb{E}[e^{\lambda(U - \mathbb{E}U)}] \leq \lambda^2 \sigma^2 / 2$ for all $\lambda \in \mathbb{R}$.

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
