[Supplementary Material]

# A Proof of Lemma 1

Just like Russo and Zou [3], we exploit the Donsker–Varadhan variational representation of the relative entropy [20, Corollary 4.15]: for any two probability measures $\pi, \rho$ on a common measurable space $(\Omega, \mathcal{F})$,

$$D(\pi\|\rho) = \sup_F \left\{ \int_\Omega F \, \mathrm{d}\pi - \log \int_\Omega e^F \mathrm{d}\rho \right\}, \tag{A.1}$$

where the supremum is over all measurable functions $F : \Omega \to \mathbb{R}$, such that $e^F \in L^1(\rho)$. From (A.1), we know that for any $\lambda \in \mathbb{R}$,

$$D(P_{X,Y}\|P_X \otimes P_Y) \geq \mathbb{E}[\lambda f(X, Y)] - \log \mathbb{E}\big[e^{\lambda f(\bar{X},\bar{Y})}\big]$$

$$\geq \lambda\big(\mathbb{E}[f(X,Y)] - \mathbb{E}[f(\bar{X},\bar{Y})]\big) - \frac{\lambda^2\sigma^2}{2}, \tag{A.2}$$

where the second step follows from the subgaussian assumption on $f(\bar{X}, \bar{Y})$:

$$\log \mathbb{E}\big[e^{\lambda(f(\bar{X},\bar{Y}) - \mathbb{E}[f(\bar{X},\bar{Y})])}\big] \leq \frac{\lambda^2\sigma^2}{2} \qquad \forall \lambda \in \mathbb{R}.$$

Inequality (A.2) gives a nonnegative parabola in $\lambda$, whose discriminant must be nonpositive, which implies

$$\big|\mathbb{E}[f(X,Y)] - \mathbb{E}[f(\bar{X},\bar{Y})]\big| \leq \sqrt{2\sigma^2 D(P_{X,Y}\|P_X \otimes P_Y)}.$$

The result follows by noting that $I(X;Y) = D(P_{X,Y}\|P_X \otimes P_Y)$.

# B Proof of Theorem 3

To prove Theorem 3, we need the following two lemmas.

**Lemma B.1.** *Consider the parallel execution of $m$ independent copies of $P_{W|S}$ on independent datasets $S_1, \ldots, S_m$: for $t = 1, \ldots, m$, an independent copy of $P_{W|S}$ takes $S_t \sim \mu^{\otimes n}$ as input and outputs $W_t$. Define $S^m \triangleq (S_1, \ldots, S_m)$. If under $\mu$, $P_{W|S}$ satisfies that $I(\Lambda_W(S); W) \leq \varepsilon$, then the overall algorithm $P_{W^m|S^m}$ satisfies $I(\Lambda_W(S_1), \ldots, \Lambda_W(S_m); W^m) \leq m\varepsilon$.*

*Proof.* The proof is based on the independence among $(S_t, W_t)$, $t = 1, \ldots, m$, and the chain rule of mutual information. □

**Lemma B.2.** *Let $S^m \triangleq (S_1, \ldots, S_m)$, where $S_t \sim \mu^{\otimes n}$. If an algorithm $P_{W,T,R|S^m} : \mathsf{Z}^{m \times n} \to \mathsf{W} \times [m] \times \{\pm 1\}$ satisfies $I(\Lambda_W(S_1), \ldots, \Lambda_W(S_m); W, T, R) \leq \varepsilon$, and if $\ell(w, Z)$ is $\sigma$-subgaussian for all $w \in \mathsf{W}$, then*

$$\mathbb{E}\big[R(L_{S_T}(W) - L_\mu(W))\big] \leq \sqrt{\frac{2\sigma^2\varepsilon}{n}}.$$

*Proof.* The proof is based on Lemma 1. Let $X = (\Lambda_W(S_1), \ldots, \Lambda_W(S_m))$, $Y = (W, T, R)$, and

$$f\big((\Lambda_W(s_1), \ldots, \Lambda_W(s_m)), (w, t, r)\big) = r L_{s_t}(w).$$

If $\ell(w, Z)$ is $\sigma$-subgaussian under $Z \sim \mu$ for all $w \in \mathsf{W}$, then $\frac{r}{n}\sum_{i=1}^n \ell(w, Z_{t,i})$ is $\sigma/\sqrt{n}$-subgaussian for all $w \in \mathsf{W}$, $t \in [m]$ and $r \in \{\pm 1\}$, and hence $f(X, Y)$ is $\sigma/\sqrt{n}$-subgaussian. Lemma 1 implies that

$$\mathbb{E}[RL_{S_T}(W)] - \mathbb{E}[RL_\mu(W)] \leq \sqrt{\frac{2\sigma^2 I(\Lambda_W(S_1), \ldots, \Lambda_W(S_m); W, T, R)}{n}}$$

and proves the claim. □

Note that the upper bound in Lemma B.2 does not depend on $m$. With these lemmas, we can prove Theorem 3.

*Proof of Theorem 3.* The proof is an adaptation of a "monitor technique" proposed by Bassily et al. [6]. First, let $P_{W^m|S^m}$ be the parallel execution of $m$ independent copies of $P_{W|S}$: for $t = 1, \ldots, m$, an independent copy of $P_{W|S}$ takes an independent $S_t \sim \mu^{\otimes n}$ as input and outputs $W_t$. Given $S^m$ and $W^m$, let the output of the "monitor" be a sample $(W^*, T^*, R^*)$ drawn from $\mathsf{W} \times [m] \times \{\pm 1\}$ according to

$$(T^*, R^*) = \underset{t \in [m],\, r \in \{\pm 1\}}{\arg \max}\, r\big(L_\mu(W_t) - L_{S_t}(W_t)\big) \quad \text{and} \quad W^* = W_{T^*}. \tag{B.3}$$

This gives

$$R^*\big(L_\mu(W^*) - L_{S_{T^*}}(W^*)\big) = \max_{t \in [m]} \big|L_\mu(W_t) - L_{S_t}(W_t)\big|.$$

Taking expectation on both sides, we have

$$\mathbb{E}\big[R^*\big(L_\mu(W^*) - L_{S_{T^*}}(W^*)\big)\big] = \mathbb{E}\Big[\max_{t \in [m]} \big|L_\mu(W_t) - L_{S_t}(W_t)\big|\Big]. \tag{B.4}$$

Note that conditional on $W^m$, the tuple $(W^*, T^*, R^*)$ can take only $2m$ values, which means that

$$I(\Lambda_\mathsf{W}(S_1), \ldots, \Lambda_\mathsf{W}(S_m); W^*, T^*, R^* | W^m) \leq \log(2m). \tag{B.5}$$

In addition, since $P_{W|S}$ is assumed to satisfy $I(\Lambda_\mathsf{W}(S); W) \leq \varepsilon$, Lemma B.1 implies that

$$I(\Lambda_\mathsf{W}(S_1), \ldots, \Lambda_\mathsf{W}(S_m); W^m) \leq m\varepsilon.$$

Therefore, by the chain rule of mutual information and the data processing inequality, we have

$$I(\Lambda_\mathsf{W}(S_1), \ldots, \Lambda_\mathsf{W}(S_m); W^*, T^*, R^*) \leq I(\Lambda_\mathsf{W}(S_1), \ldots, \Lambda_\mathsf{W}(S_m); W^m, W^*, T^*, R^*)$$
$$\leq m\varepsilon + \log(2m).$$

By Lemma B.2 and the assumption that $\ell(w, Z)$ is $\sigma$-subgaussian,

$$\mathbb{E}\big[R^*\big(L_{S_{T^*}}(W^*) - L_\mu(W^*)\big)\big] \leq \sqrt{\frac{2\sigma^2}{n}\big(m\varepsilon + \log(2m)\big)}. \tag{B.6}$$

Combining (B.6) and (B.4) gives

$$\mathbb{E}\Big[\max_{t \in [m]} \big|L_{S_t}(W_t) - L_\mu(W_t)\big|\Big] \leq \sqrt{\frac{2\sigma^2}{n}\big(m\varepsilon + \log(2m)\big)}. \tag{B.7}$$

The rest of the proof is by contradiction. Choose $m = \lfloor 1/\beta \rfloor$. Suppose the algorithm $P_{W|S}$ does not satisfy the claimed generalization property, namely,

$$\mathbb{P}\big[\big|L_S(W) - L_\mu(W)\big| > \alpha\big] > \beta. \tag{B.8}$$

Then by the independence among the pairs $(S_t, W_t)$, $t = 1, \ldots, m$,

$$\mathbb{P}\Big[\max_{t \in [m]} \big|L_{S_t}(W_t) - L_\mu(W_t)\big| > \alpha\Big] > 1 - (1 - \beta)^{\lfloor 1/\beta \rfloor} > \frac{1}{2}.$$

Thus

$$\mathbb{E}\Big[\max_{t \in [m]} \big|L_{S_t}(W_t) - L_\mu(W_t)\big|\Big] > \frac{\alpha}{2}. \tag{B.9}$$

Combining (B.7) and (B.9) gives

$$\frac{\alpha}{2} < \sqrt{\frac{2\sigma^2}{n}\Big(\frac{\varepsilon}{\beta} + \log\frac{2}{\beta}\Big)}. \tag{B.10}$$

The above inequality implies that

$$n < \frac{8\sigma^2}{\alpha^2}\Big(\frac{\varepsilon}{\beta} + \log\frac{2}{\beta}\Big), \tag{B.11}$$

which contradicts the condition in (16). Therefore, under the condition in (16), the assumption in (B.8) cannot hold. This completes the proof. $\qquad \square$

## C  Proof of Theorem 5

To solve the relaxed optimization problem in (26), first note that

$$\inf_{P_{W|S}} \left( \mathbb{E}[L_S(W)] + \frac{1}{\beta} D(P_{W|S} \| Q | P_S) \right)$$

$$= \inf_{P_{W|S}} \int_{\mathsf{Z}^n} \mu^{\otimes n}(\mathrm{d}s) \left( \mathbb{E}[L_s(W)|S=s] + \frac{1}{\beta} D(P_{W|S=s} \| Q) \right)$$

$$= \int_{\mathsf{Z}^n} \mu^{\otimes n}(\mathrm{d}s) \inf_{P_{W|S=s}} \left( \mathbb{E}[L_s(W)|S=s] + \frac{1}{\beta} D(P_{W|S=s} \| Q) \right).$$

It follows that for each $s \in \mathsf{Z}^n$, the algorithm $P^*_{W|S}$ that minimizes (26) satisfies

$$P^*_{W|S=s} = \arg\inf_{P_{W|S=s}} \left( \mathbb{E}[L_s(W)|S=s] + \frac{1}{\beta} D(P_{W|S=s} \| Q) \right). \tag{C.12}$$

This is a simple convex optimization problem. The solution to (C.12) for each $s \in \mathsf{Z}^n$ turns out to be the Gibbs algorithm [21] as described in (27), which does not depend on $\mu$.

## D  Proof of Corollary 2

We can bound the expected empirical risk of the Gibbs algorithm $P^*_{W|S}$ as

$$\mathbb{E}[L_S(W)] \le \mathbb{E}[L_S(W)] + \frac{1}{\beta} D(P^*_{W|S} \| Q | P_S) \tag{D.13}$$

$$\le \mathbb{E}[L_S(w)] + \frac{1}{\beta} D(\delta_w \| Q) \qquad \text{for all } w \in \mathsf{W}, \tag{D.14}$$

where $\delta_w$ is the point mass at $w$. The second inequality is due to Theorem 5, as $\delta_w$ can be viewed as a learning algorithm that ignores the dataset and always outputs $w$. Taking $w = w_\mathrm{o}$, noting that $\mathbb{E}[L_S(w_\mathrm{o})] = L_\mu(w_\mathrm{o})$, and combining with the upper bound on the expected generalization error (28), we obtain

$$\mathbb{E}[L_\mu(W)] \le \inf_{w \in \mathsf{W}} L_\mu(w) + \frac{1}{\beta} D(\delta_{w_\mathrm{o}} \| Q) + \frac{\beta}{2n}. \tag{D.15}$$

This leads to (29), as $D(\delta_{w_\mathrm{o}} \| Q) = -\log Q(w_\mathrm{o})$ when $\mathsf{W}$ is countable.

## E  Proof of Corollary 3

Similar to the proof of Corollary 2, we first bound the expected empirical risk of the Gibbs algorithm $P^*_{W|S}$. For any $a > 0$, $\mathcal{N}(w_\mathrm{o}, a^2 \mathbf{I}_d)$ can be viewed as a learning algorithm that ignores the dataset and always draws a hypothesis from this distribution. The nonnegativity of relative entropy and Theorem 5 imply that

$$\mathbb{E}[L_S(W)] \le \mathbb{E}[L_S(W)] + \frac{1}{\beta} D(P^*_{W|S} \| Q | P_S) \tag{E.16}$$

$$\le \int_{\mathsf{W}} \mathbb{E}[L_S(w)] \mathcal{N}(w; w_\mathrm{o}, a^2 \mathbf{I}_d) \mathrm{d}w + \frac{1}{\beta} D\big(\mathcal{N}(w_\mathrm{o}, a^2 \mathbf{I}_d) \| Q\big) \tag{E.17}$$

$$= \int_{\mathsf{W}} L_\mu(w) \mathcal{N}(w; w_\mathrm{o}, a^2 \mathbf{I}_d) \mathrm{d}w + \frac{1}{\beta} D\big(\mathcal{N}(w_\mathrm{o}, a^2 \mathbf{I}_d) \| Q\big). \tag{E.18}$$

Combining with the upper bound on the expected generalization error (28), we obtain

$$\mathbb{E}[L_\mu(W)] \le \inf_{a > 0} \left( \int_{\mathsf{W}} L_\mu(w) \mathcal{N}(w; w_\mathrm{o}, a^2 \mathbf{I}_d) \mathrm{d}w + \frac{1}{\beta} D\big(\mathcal{N}(w_\mathrm{o}, a^2 \mathbf{I}_d) \| Q\big) \right) + \frac{\beta}{2n}. \tag{E.19}$$

Since $\ell(\cdot, z)$ is $\rho$-Lipschitz for all $z \in \mathsf{Z}$, we have that for any $w \in \mathsf{W}$,

$$|L_\mu(w) - L_\mu(w_\mathrm{o})| \le \mathbb{E}[|\ell(w, Z) - \ell(w_\mathrm{o}, Z)|] \le \rho \|w - w_\mathrm{o}\|. \tag{E.20}$$

Then

$$\int_{\mathsf{W}} L_\mu(w)\mathcal{N}(w; w_\mathrm{o}, a^2\mathbf{I}_d)\mathrm{d}w \le \int_{\mathsf{W}} \big(L_\mu(w_\mathrm{o}) + \rho\|w - w_\mathrm{o}\|\big)\mathcal{N}(w; w_\mathrm{o}, a^2\mathbf{I}_d)\mathrm{d}w \qquad \text{(E.21)}$$

$$\le L_\mu(w_\mathrm{o}) + \rho a\sqrt{d}. \qquad \text{(E.22)}$$

Substituting this into (E.19), we obtain (31).

# F Proof of Corollary 4

We prove the result assuming $|\mathsf{W}| = k$. When $\mathsf{W}$ is countably infinite, the proof carries over by replacing $k$ with $\infty$.

First, we upper-bound the expected generalization error via $I(S; W)$. We have the following chain of inequalities:

$$I(S; W) \le I\big((L_S(w_i))_{i\in[k]}; (L_S(w_i) + N_i)_{i\in[k]}\big) \qquad \text{(F.23)}$$

$$\le \sum_{i=1}^{k} I(L_S(w_i); L_S(w_i) + N_i) \qquad \text{(F.24)}$$

$$\le \sum_{i=1}^{k} \log\left(1 + \frac{\mathbb{E}[L_S(w_i)]}{b_i}\right) \qquad \text{(F.25)}$$

$$= \sum_{i=1}^{k} \log\left(1 + \frac{L_\mu(w_i)}{b_i}\right), \qquad \text{(F.26)}$$

where we have used the data processing inequality for mutual information; the fact that for product channels, the mutual information between the overall input and output is upper-bounded by the sum of the input-output mutual information of individual channels [22]; the formula for the capacity of the additive exponential noise channel under an input mean constraint [23]; and the fact that $\mathbb{E}[L_S(w_i)] = L_\mu(w_i)$. The assumption that $\ell$ takes values in $[0, 1]$ implies that $\ell(w, Z)$ is $1/2$-subgaussian for all $w \in \mathsf{W}$, and as a consequence of (F.26),

$$\mathrm{gen}(\mu, P_{W|S}) \le \sqrt{\frac{1}{2n}\sum_{i=1}^{k}\log\left(1 + \frac{L_\mu(w_i)}{b_i}\right)}. \qquad \text{(F.27)}$$

Then, we upper-bound the expected empirical risk. From the definition of the algorithm, we have that with probability one,

$$L_S(W) = L_S(W) + N_W - N_W \qquad \text{(F.28)}$$

$$\le L_S(w_{i_\mathrm{o}}) + N_{i_\mathrm{o}} - N_W \qquad \text{(F.29)}$$

$$\le L_S(w_{i_\mathrm{o}}) + N_{i_\mathrm{o}} - \min\{N_i, i \in [k]\}. \qquad \text{(F.30)}$$

Taking expectation on both sides, we get

$$\mathbb{E}[L_S(W)] \le L_\mu(w_{i_\mathrm{o}}) + b_{i_\mathrm{o}} - \left(\sum_{i=1}^{k}\frac{1}{b_i}\right)^{-1}. \qquad \text{(F.31)}$$

Combining (F.27) and (F.31), we have

$$\mathbb{E}[L_\mu(W)] \le \min_{i\in[k]} L_\mu(w_i) + \sqrt{\frac{1}{2n}\sum_{i=1}^{k}\log\left(1 + \frac{L_\mu(w_i)}{b_i}\right)} + b_{i_\mathrm{o}} - \left(\sum_{i=1}^{k}\frac{1}{b_i}\right)^{-1}, \qquad \text{(F.32)}$$

which leads to (34) with the fact that $\log(1 + x) \le x$.

When $b_i = i^{1.1}/n^{1/3}$, using the fact that

$$\sum_{i=1}^{k}\frac{1}{i^{1.1}} \le 11 - 10k^{-1/10} \qquad \text{(F.33)}$$

and upper-bounding $L_\mu(w_i)$'s by 1, we get

$$\mathbb{E}[L_\mu(W)] \leq \min_{i \in [k]} L_\mu(w_i) + \frac{1}{n^{1/3}} \left( \sqrt{\frac{1}{2} \left( 11 - 10k^{-1/10} \right)} + i_{\mathrm{o}}^{1.1} - \frac{1}{11 - 10k^{-1/10}} \right) \quad \text{(F.34)}$$

$$\leq \min_{i \in [k]} L_\mu(w_i) + \frac{3 + i_{\mathrm{o}}^{1.1}}{n^{1/3}}, \quad \text{(F.35)}$$

which proves (35).