[Reviews · NeurIPS 2017]

Reviewer 1



This paper follows up on the line of work introduced by Russo and Zou, that uses information theoretic bounds to provide bounds for adaptive data analysis. Here, the authors focus on learning algorithms. The authors use the input-output mutual information instead of the collection of empirical risks as in Russo—applied on adaptive composition of learning algorithms, this later allows for stronger connections to the hypothesis space’s VC dimension. The paper was a pleasure to read with only a single grievance regarding typography (below under ‘minor points’). The work was scoped carefully and clearly in the context of prior work. The results are intersting: they tighten some known analysis, but also provide new, high probability results. A nice part of the paper is where the authors use the info theoretic framework to demonstrate that the Gibbs algorithm effectively is ERM plus a (upper bound on the) regularizer of the input-output mutual information. Similarly, they motivate noisy ERM as a way to control generalization error. This last, algorithmic part is what got me the most excited, and left me wanting more. I would like to see more prescriptive discussion and methodology. How can this translate into instructions for practitioners? Even a discussion of open problems/questions in that direction would be good to have. Minor points: - A bit confusing that the authors use uppercase W both for the hypothesis space and the output hypothesis. I can see that the typefaces are different, but I feel that they’re not different enough to help me read this more smoothly.

Reviewer 2



This paper derives upper bounds on the generalization error of learning algorithms in terms of the mutual information between the data S and the hypothesis W. Using Lemma 1, their first result is to show that the generalization error can be upper-bounded by a quantity that is proportional to the square root of the mutual information between S and W, as long as the loss is \sigma-subgaussian. Lemma 1 also enables the authors to immediately recover a previous result by Russo and Zou. They furthermore provide a high-probability bound and an expected bound on the absolute difference between the population risk and the empirical risk. The next section then applies these bounds to specific algorithms: two of them have an hypothesis set that can be easily upper bounded; the next two have a bounded mutual info between S and W that can be upper bounded; then the authors discuss practical methods to effectively bound I(S;W); and finally, the generalization error of a composite learning algorithm is analyzed. Finally, a similar analysis is then employed to “adaptive data analytics”. PROS: - The authors attempt to bridge information-theoretic ideas with learning theory. This is something that I believe would enormously favor both communities. - The first bounds on the generalization error based on the mutual information are intuitively appealing. (It is however unclear to me how novel they are relative to Russo and Zou's prior work). CONS: - The paper is very dense as it is, and would have benefited from more concrete, lively examples that the NIPS community can relate to in order to illustrate the results. In particular, I feel it's a terrible shame that most likely only minute fraction of the NIPS community will find interest in the results of this paper, just because of the way it is written. This is the main reason why I felt obliged to decrease my score. - All the results seem to require the sigma-subgaussianity of the loss. How restrictive is this? Can you give concrete examples where the analysis holds and where it fails?

Reviewer 3



Authors provide upper bounds on the generalization error of supervised learning algorithms via the mutual information between the chosen hypothesis class and the sample set. These bounds can be further used to recover the VC dimension bounds up to a multiplicative factor of sqrt(log n). The approach provides an alternate proof technique to the traditional Radamacher complexity bounds for binary classification. I enjoyed reading the paper and I recommend acceptance.